# Genes on the Move: In Vitro Transduction of Antimicrobial Resistance Genes between Human and Canine Staphylococcal Pathogens

**DOI:** 10.3390/microorganisms8122031

**Published:** 2020-12-18

**Authors:** Sian Marie Frosini, Ross Bond, Alex J. McCarthy, Claudia Feudi, Stefan Schwarz, Jodi A. Lindsay, Anette Loeffler

**Affiliations:** 1Department of Clinical Science and Services, Royal Veterinary College, Hawkshead Lane, North Mymms, Hatfield, Hertfordshire AL9 7TA, UK; rbond@rvc.ac.uk (R.B.); aloeffler@rvc.ac.uk (A.L.); 2MRC Centre for Molecular Bacteriology and Infection, Imperial College London, London SW7 2AZ, UK; a.mccarthy@imperial.ac.uk; 3Centre for Infection Medicine, Department of Veterinary Medicine, Institute of Microbiology and Epizootics, Freie Universität Berlin, 14195 Berlin, Germany; Claudia.Feudi@fu-berlin.de (C.F.); Stefan.Schwarz@fu-berlin.de (S.S.); 4Institute of Infection and Immunity, St George’s, University of London, Cranmer Terrace, London SW17 0RE, UK; jlindsay@sgul.ac.uk

**Keywords:** staphylococci, zoonosis, MRSA, bacteriophage, MRSP

## Abstract

Transmission of methicillin-resistant *Staphylococcus aureus* (MRSA) and methicillin-resistant *Staphylococcus pseudintermedius* (MRSP) between people and pets, and their co-carriage, are well-described. Potential exchange of antimicrobial resistance (AMR) genes amongst these staphylococci was investigated in vitro through endogenous bacteriophage-mediated transduction. Bacteriophages were UV-induced from seven donor isolates of canine (MRSP) and human (MRSA) origin, containing *tet*(M), *tet*(K), *fusB* or *fusC*, and lysates filtered. Twenty-seven tetracycline- and fusidic acid- (FA-) susceptible recipients were used in 122 donor-recipient combinations (22 tetracycline, 100 FA) across 415 assays (115 tetracycline, 300 FA). Bacteriophage lysates were incubated with recipients and presumed transductants quantified on antimicrobial-supplemented agar plates. Tetracycline resistance transduction from MRSP and MRSA to methicillin-susceptible *S. pseudintermedius* (MSSP) was confirmed by PCR in 15/115 assays. No FA-resistance transfer occurred, confirmed by negative *fusB/fusC* PCR, but colonies resulting from FA assays had high MICs (≥32 mg/L) and showed mutations in *fusA*, two at a novel position (F88L), nine at H457[Y/N/L]. Horizontal gene transfer of tetracycline-resistance confirms that resistance genes can be shared between coagulase-positive staphylococci from different hosts. Cross-species AMR transmission highlights the importance of good antimicrobial stewardship across humans and veterinary species to support One Health.

## 1. Introduction

Transmission of multidrug-resistant (MDR) bacterial pathogens between humans and pets contributes to the spread of antimicrobial resistance (AMR) and is facilitated by frequent close contact and advanced veterinary care [1]. While the transfer of MDR bacteria between hosts can be mitigated through screening and hygiene measures, transfer of resistance determinants between co-colonising bacteria will follow microbial rules of gene exchange.

Methicillin-resistant *Staphylococcus aureus* (MRSA) presents a significant burden to human healthcare through poorer clinical outcomes and higher treatment costs compared with methicillin-susceptible *S. aureus* (MSSA) [2,3]. MRSA is occasionally isolated from infections in pets, typically after reverse zoonotic transmission [4]. More recently, though, its “canine counterpart”, methicillin-resistant *S. pseudintermedius* (MRSP), has emerged as a highly drug-resistant, zoonotic pathogen in veterinary clinics [5,6]. Although MRSP is primarily adapted to dogs, it shares many microbiological, clinical and epidemiological characteristics with MRSA. Both are coagulase-positive opportunistic pathogens with the ability to colonise mucosae and skin asymptomatically. Simultaneous co-carriage of and infection with *S. aureus* and *S. pseudintermedius* have been documented in humans and dogs [7,8,9].

The acquisition or loss of mobile genetic elements (MGEs) carrying AMR genes, including plasmids, transposons and staphylococcal cassette chromosome *mec* (SCC*mec*) elements, can lead to phenotypic changes in AMR profiles of staphylococci [10]. Horizontal gene transfer (HGT) of MGEs can occur between individual bacteria by transformation, conjugation, or transduction [11]. In *S. aureus*, this is thought to be primarily by bacteriophage-mediated generalised transduction [12]. Comparatively little information exists for *S. pseudintermedius* [13], but transduction seems the most likely mechanism for HGT amongst co-colonising isolates. Integrated bacteriophages (prophages) have been identified in *S. pseudintermedius* chromosomes while the *tra* gene complex, required for conjugation, was not found in 15 sequenced isolates [14,15]. Transformation, which does not require cell-to-cell contact, appears to occur rarely in *S. aureus* under natural conditions [12].

Bacteriophage-mediated generalised transduction relies on the presence of bacteriophage receptors in recipient bacteria and is further dependent on the ability of MGEs to replicate or integrate into the new host’s genome. HGT is controlled by Restriction-Modification (RM) systems and, more rarely in staphylococci, by clusters of randomly interspersed short palindromic repeats (CRISPR) systems, which protect bacteria from acquiring foreign DNA [12]. The distribution of RM variants is lineage-associated in both *S. aureus* [16] and *S. pseudintermedius* [15], resulting in different MGEs circulating within distinct *S. aureus* or *S. pseudintermedius* lineages.

Evidence for endogenous inter-species HGT of resistance determinants in staphylococci is currently limited to transfer from coagulase-negative species (CoNS) or enterococci to *S. aureus* [17,18]. Phenotypic resistance to gentamicin, tetracycline and erythromycin has previously been transferred in vitro and on mouse skin from *S. hominis* and *S. epidermidis* into *S. aureus* (both human- and canine-derived) [17]. Moreover, the large MGE SCC*mec*, responsible for broad β-lactam resistance in MRSA and MRSP, is thought to have been transferred from CoNS [19,20]. The rapid accumulation of multiple resistance genes in MRSP suggests a less restrained acquisition of genetic material. In vivo, unexpectedly high transfer rates of MGEs, containing genes related to host-adaptation, have been observed in co-colonising *S. aureus* [21]. This is thought to be resulting from stress-linked generalised transduction [21].

Almost all clinically relevant antimicrobial classes in human medicine are also authorised and used globally in small animal veterinary practice [22,23]. One of the antimicrobial agents reserved for the treatment of serious infections caused by MRSA in humans is fusidic acid (FA) which is also widely used topically in dogs for the treatment of ear, eye and skin infections [24]. “Low-level” (Minimum Inhibitory Concentration [MIC] 4–16 mg/L) FA resistance in both *S. aureus* and *S. pseudintermedius* has been associated with *fusB* or *fusC* [25,26]. These genes have been primarily described on transposon- or SCC*mec*-like elements, found within plasmids, staphylococcal pathogenicity islands (SaPIs), or being chromosomally integrated [25,26]. Whether these MGEs can transfer between *S. aureus* and *S. pseudintermedius* remains to be answered. High-level resistance to FA (MIC ≥ 64 mg/L) has been linked to chromosomal mutations (in *fusA* and/or *fusE* in small colony variants) [25]. Another antimicrobial agent of importance in human and veterinary medicine is tetracycline, a broad-spectrum agent classified by the WHO as “highly important” for humans [22] and widely used for the treatment of respiratory tract infections in animals [27]. However, the wide distribution of tetracycline resistance genes, and their location on transposons (e.g., Tn*916*) and plasmids [28], suggests a propensity for HGT, evidence for which has yet to be shown.

In this study, we demonstrate HGT of resistance genes between isolates of *S. aureus* and *S. pseudintermediu*s using assays to detect transduction mediated by induction of natural bacteriophages.

## 2. Materials and Methods

### 2.1. Bacterial Isolates

A total of seven donor and 27 recipient bacterial isolates were used from a frozen archive (−20 °C in brain heart infusion broth (BHIB; Oxoid, Basingstoke, UK) and 20% glycerol (Fisher Scientific, Loughborough, UK) (Table 1). Selection criteria were their tetracycline and FA resistance phenotypes (disk diffusion for tetracycline, MICs for FA), genotypes, and their isolation sites, to span human, canine, infection and carriage origins and a range of sequence types (STs). All *S. pseudintermedius* isolates were collected from clinical submissions, representing the circulating lineages at the time (2007 [*n* = 1], 2010–2016 [*n* = 20]). MRSA isolates (CC8 and CC22) represented two clonal complexes found worldwide [29]. Species and respective resistances were confirmed by PCR following previously described methods [30,31] for species-specific thermonuclease (*nuc*), methicillin-resistance (*mecA*), and presence or absence of *tet(M), tet(K), fusB,* and *fusC*.

Donor isolates for tetracycline assays comprised one well-characterised MRSA of human infection origin (COL), carrying *tet(K)* on plasmid *p*T181, and one fully sequenced, prophage-positive canine infection MRSP (1726) with *tet(M)* on Tn*916* [15]. Donor isolates for FA experiments included two *fusB*-positive and three *fusC*-positive MRSP, with resistance genes most likely on transposon-like elements in plasmids (*fusB*) or integrated into the chromosomal DNA in a SCC*mec*-like cassette (*fusC*). Selection of FA-resistant donors was limited by the infrequent description of these genes in this species [30]; FA-resistant *S. aureus* donors were not available for inclusion at the time. Recipient bacteria representing different origins and STs were chosen; all were screened on brain heart infusion agar (BHIA; Oxoid) containing either 30 mg/L tetracycline or 16 mg/L FA to confirm phenotypic susceptibility. Two RM-deficient *S. aureus* laboratory strains were included as hyper-receptive recipient isolates [18].

To investigate the acquisition of tetracycline resistance, 22 different combinations of two donors and 14 recipients, including the combination of MRSA COL and RM-deficient *S. aureus* RN4220 were used; for FA assays, 100 combinations of five donors and 20 recipients were performed (Table 1). Initially, all transduction assays were performed in triplicate, but for successful combinations (confirmed by PCR for resistance gene in putative transductants), a further seven experiments (total ten replicates) were performed.

### 2.2. Induction of Bacteriophage

Overnight colonies from pure culture were grown in BHIB at 37 °C with shaking for 3 h; 1 mL aliquots were centrifuged (3000 × *g*, 3 min), and supernatant discarded. Cell pellets were resuspended in 7 mL bacteriophage buffer (0.1% 1M MgSO_4_, 0.4% CaCl_2_, 5% 1M Tris-HCl pH 7.8, 0.59% NaCl, 0.1% gelatin; Sigma-Aldrich Ltd., Gillingham, UK) and transferred to Petri dishes. The open Petri dish was exposed to UV light (302 nm, UVP Dual-Intensity Transilluminator TM-20) for 20 s to induce prophages [32,33]. Dish contents were added to 7 mL BHIB, incubated for 10 min at room temperature, then for 2 h at 32 °C with gentle agitation, and finally overnight at room temperature to allow cell lysis. Lysates were filtered (0.22 µm filter) and kept at 4 °C before being used for replicate experiments.

### 2.3. Bacteriophage Count

Recipient RN4220 colonies were incubated in 20 mL BHIB at 37 °C with shaking for 3 h. Bacteriophage lysate was diluted in bacteriophage buffer (10^−1^ and 10^−2^); 100 μL of each was added to 400 μL recipient cell broth and 30 μL 1M CaCl_2_ and incubated at room temperature for 15 min. Dilutions were mixed with 7 mL bacteriophage top agar (bacteriophage buffer containing 2 mg/L agar), poured over bacteriophage bottom agar plates (10 mg/L agar) and incubated at 32 °C for 24 h. Number of lysis plaques within the bacterial lawn were counted, with one plaque representing one phage particle.

### 2.4. Bacteriophage Transduction

Recipient bacteria were incubated in 20 mL LK broth (LKB; Luria broth with KCl instead of NaCl; 1% tryptone, 0.5% yeast extract, 0.7% KCl; Sigma-Aldrich Ltd., Gillingham, UK) at 37 °C overnight with shaking. Broth was centrifuged (4000× *g*, 10 min), supernatant discarded, and cell pellets resuspended in 1 mL LKB. In total, 100 µL of the recipient cell suspension, 100 µL bacteriophage lysate, and 200 µL LKB along with 2 µL CaCl_2_ (Sigma-Aldrich Ltd., Gillingham, UK; to a final concentration of 8 mM) were incubated at 37 °C for 45 min with shaking. Subsequently, 200 µL ice-cold 0.02 M sodium citrate was added (Honeywell International Inc., Bucharest, Romania) to chelate calcium and prevent further phage binding and cell lysis. Cell suspensions were centrifuged (3000× *g*, 3 min), supernatant discarded, the pellet resuspended in 200 µL ice-cold sodium citrate, and left for 2 h on ice [33].

The 200 µL solutions were spread using hockey-stick spreaders onto the surface of an LK bottom agar plate (10 g/L agar) containing sub-inhibitory antimicrobial concentrations to induce resistance gene expression (0.3 mg/L tetracycline or 0.03 mg/L FA) and incubated at 37 °C for 45 min. Four-to-five mL of LK top agar (2 g/L agar) containing inhibitory antimicrobial concentrations (30 mg/L tetracycline in total or 16 mg/L FA in total) were overlaid, plates incubated upright for 48 h at 37 °C, and colonies counted.

A negative control with 100 µL LKB in place of bacteriophage lysate was included for every combination and growth compared to transduction plates. Colony numbers at least twice those seen on the corresponding negative control were deemed significant growth, indicative of resistance transfer.

### 2.5. Confirmation of Suspected Transductants

From each assay with significant growth, 2–9 putative transductant colonies were subcultured onto BHIA containing either 30 mg/L tetracycline or 16 mg/L FA to confirm phenotypic susceptibility; expected species and the presence/absence of respective resistance genes were again investigated [30,31]. For isolates grown on FA-supplemented agar but negative for *fusB* and *fusC*, MICs were determined for at least two colonies, as well as for their respective donor and recipient [30]. In 1 to 3 representative *fusB/fusC* negative post-transduction colonies from each recipient with MICs ≥ 32 mg/L, *fusA* was amplified and sequenced alongside that of their original recipient following a previously described method [30].

### 2.6. Statistical Analyses

In IBM SPSS Statistics version 26 (significance *p* < 0.05), transduction rates (transductants/mL) and frequencies were compared by Kruskal-Wallis tests with the Dunn-Bonferroni post hoc method.

## 3. Results

### 3.1. Bacteriophage Count

Bacteriophage count could not be established as the RN4220 bacterial lawn did not show any lytic plaques for phage lysate from any donor; transducing phage counts have been shown previously not to correlate with lytic phage counts [33].

### 3.2. Transduction of Tetracycline Resistance

To study HGT of tetracycline resistance, bacteriophage lysates from one *tet(M)*-positive and one *tet(K)*-positive donor were cultured with 14 tetracycline-susceptible recipients. Phenotypically tetracycline-resistant colonies grew from seven of the 22 different donor/recipient combinations (initially done in triplicate) (Table 1); expected nuc and acquisition of *tet(M)* or *tet(K)* were confirmed in all. Transfer occurred from MRSA COL into control MSSA RN4220 and three methicillin-susceptible *S. pseudintermedius* (MSSP) recipients, and from MRSP 1726 into three MSSP recipients (Figure 1). In contrast, no transduction of phenotypic tetracycline resistance was seen from MRSP into *S. aureus* (including both RM-deficient recipients). Including the subsequent additional seven replicates from successful pairings (115 assays in total), transduction occurred in 15/115 assays, confirmed by PCR in all 38 tested colonies. Reproducibility was low in most replicate experiments with a maximum of 4/10 positive repeats found from MRSA COL into RN4220 and from MRSP into an MSSP. Growth of <10 colonies per plate (Figure 1) was seen on 9/23 negative controls, representing seven recipients (6 MSSP, 1 MSSA). There was no difference (*P* = 0.994) between colony counts/mL for transduction between MRSP-MSSP, MRSA-MSSP or MRSA-RM-deficient MSSA (Table 2).

### 3.3. Transduction of FA Resistance

For FA resistance, bacteriophage lysate from two *fusB*- and three *fusC*-positive MRSP donors was combined with 20 FA-susceptible recipients (Table 1). Of the 300 transduction plates in total, 18 showed significant growth. Growth of up to 50 colonies was seen on 24/35 negative control plates from all but three (P1361, V1273, B021) recipients. Neither *fusB* nor *fusC* were detected in the 59 colonies tested post-transduction. Significant growth was seen more frequently on transduction assays for MRSA recipients (13/120 plates) than for MSSP recipients (2/120 plates; *P* = 0.032); the frequency of growth was similar for the 60 MSSA assays.

All tested colonies from FA transduction assays (two from each plate) had MICs higher than their donor (donors 4 mg/L–16 mg/L; putative transductants 32 mg/L–>64 mg/L) and their recipient isolates (0.03 mg/L–0.06 mg/L). Sequencing of *fusA* in 11/11 post-transduction assay colonies identified mutations in one of two amino acid positions (Table 3). The most common mutation (9/11 colonies) was in amino acid 457 (H457Y, H457N, H457L); two colonies had the mutation F88L, located in domain I of *fusA*.

## 4. Discussion

For the first time, our results provide phenotypic and molecular evidence for AMR transfer between different coagulase-positive staphylococcal species from human and canine origin, mediated by endogenous bacteriophages.

This cross-species spread of AMR, from the human pathogen *S. aureus* into the canine pathogen *S. pseudintermedius*, is of particular relevance to the often-close contact settings between pet owners and their pets, with *S. aureus* acting as a potential reservoir of resistance genes for *S. pseudintermedius*. It draws new attention to a potential risk to pets from contact with humans. This adds to a wealth of information focusing on the irrefutable priority direction of pet-to-human transfer [34]. Dogs and humans may be at least transient carriers (and co-carriers) of staphylococcal species adapted to the respective “other” primary host [4,5]. Our results add an extra layer of complexity to the potential clinical implications of close companionship with our pets, should HGT occur from *S. aureus* to *S. pseudintermedius* in vivo. Why no transduction of tetracycline resistance genes occurred from MRSP into *S. aureus* (including RN4220) remains unclear, but may include more efficient RM-systems, CRISPRs (although rarely described in staphylococci), a lack of bacteriophage receptors in *S. aureus*, plasmid incompatibility, or non-compatible RM systems (which may or may not be lineage specific) [12,14,16]. Similar unilateral transfer preferences were previously noted in an earlier study using exogenous bacteriophages in other staphylococcal species [35], although HGT was observed bidirectionally between *S. aureus* and *S. pseudintermedius*.

The low reproducibility of transduction of tetracycline resistance genes in successful pairings was surprising. It may have been due to low concentrations of endogenous transducing bacteriophages in lysates, or low copy number of *tet(M)*/*tet(K)* within induced bacteriophages. While this may suggest that cross-species gene exchange represents only a minor contribution to the overall spread of AMR, our findings prove a new concept in the evolution of MDR pathogens, in an area directly impacting on human health. Furthermore, transduction rates are difficult to compare as the number of successful replicates are rarely stated (instead described as variation (mean ± SD)). Our transduction rates in successful replicates (number of transductant cells/mL) were similar to those described previously using UV-light induction of bacteriophages (approximately 10–350 cfu/mL previously c.f. 25–1535 in this study; Table 2) [33]. Two MSSP recipients (221833 and 287735) had particularly high transductant cell counts, suggesting they may have weaker transfer barriers or greater phage receptor expression, allowing a higher transduction rate. This is also indicated by the acceptance of DNA by recipients 221833 and 289595 in more replicate experiments, from both MRSA and MRSP donors. It is possible that transfer of resistance genes via transformation of DNA present in lysates could occur, however *S. aureus* competence genes are poorly expressed by mutated sigma factors and post-transcriptional control, resulting in extremely rare transfer frequency [12,36]. The reasons why the phage lysate did not form plaques in the RN4220 bacterial lawn are unclear. Potentially this could be due to missing or modified phage receptors in this strain, or the induction of a novel transducing phage. It is possible that despite being RM-deficient to our current knowledge, RN4220 may contain other undiscovered types of phage immunity. This non-plaque-forming phenomenon with RN4220 is not uncommon to see when plating transducing phages induced from clinical *S. aureus* isolates (unpublished data), and it has been previously demonstrated that the presence of lytic phages does not correlate with transducing phage [33]. However, it cannot be discounted that the apparent absence of FA resistance gene transduction could be the result of a lack of transducing phage.

The risk of interspecies HGT may be greater in vivo than in the laboratory, as has been demonstrated previously for other MGEs [17,21]. Plasmid-borne gentamicin resistance transfer from the coagulase-negative *S. epidermidis* into *S. aureus* was 10-100-fold greater on mouse skin than in broth filter experiments [17] and similarly, HGT of host-adaptation determinants on pig skin was substantially higher compared to the same isolates co-incubated in vitro [21]. However, the reasons underpinning why transfer is observed at a higher rate in vivo than in vitro are still largely unknown. It is thought that this may relate to environmental conditions that are not replicated in vitro, which may trigger staphylococcal isolates to selectively amplify HGT.

The lack of transfer of *fusB* and *fusC* in this study is encouraging with regard to the preservation of FA clinical efficacy in human and veterinary medicine. However, the finding needs to be interpreted with caution. Firstly, the development of high-level resistance likely due to *fusA* mutations following exposure to relatively low concentrations of FA is of concern, although *fusA* mutations are rarely documented in clinical isolates [25,30]. This low prevalence of FA resistance, despite FA being widely used in veterinary and human medicine for over 50 years, suggests that its use is not causing a ‘crisis’ of resistance. Indeed, in veterinary medicine FA is used as topical therapy where it exceeds typical MICs for staphylococci by a significant order of magnitude [36]. Thus, it seems prudent to suggest that proactive surveillance of resistance in both human- and veterinary-derived staphylococci would suffice to monitor this situation. However, it does not indicate a current need to restrict the use of this antimicrobial to humans only at this time. In this study, transduction may have also been hampered by a lack of prophage in our donors, as reported for a small number of *S*. *pseudintermedius* lineages [14]. The mutations observed in *fusA* of *S. pseudintermedius* occurred in the same position as described in *S. aureus* (amino acid 457), confirming the importance of this mutation in resistance development [25]. The role of the novel mutation (F88L) in conferring tolerance to FA should be further investigated.

In conclusion, the description of MGE transfer between *S. aureus* and *S. pseudintermedius* illustrates ongoing genetic evolution amongst major zoonotic staphylococcal pathogens. Selective pressures in one host may thus contribute to the evolution of more drug-resistant isolates adapted to another host. Whilst the wider context of direction of transfer and the prioritisation of human over animal health remain important considerations, there is clearly a need for response to the dissemination of AMR within shared bacterial populations. Despite previous significant attention on the use of antimicrobial agents in livestock, companion animal medicine is in some ways left lagging behind. Efforts to develop and disseminate responsible antimicrobial use guidelines for companion animal medicine need to continue, also to align interests in the sense of One Health. At present, though, the well-documented benefit from pet ownership on human health likely markedly outweighs the risk from zoonotic transmission and HGT in methicillin-resistant staphylococci [37].

## Figures and Tables

**Figure 1 microorganisms-08-02031-f001:**
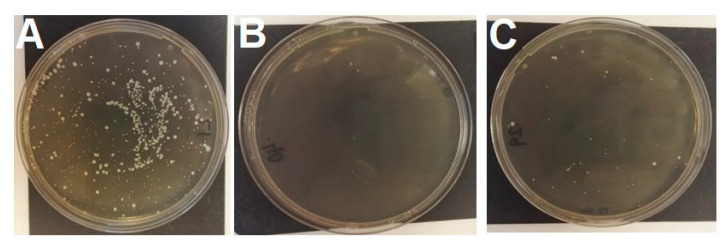
Recipient MSSP 287735 growth after transduction assays (**A**) on agar containing 30 mg/L tetracycline with phage lysate from MRSP 1726 (*tet(M)* donor); (**B**) on agar containing 30 mg/L tetracycline control with no phage lysate; (**C**) on agar containing 16 mg/L fusidic acid with no phage lysate. Note the breakthrough growth on plate C (colony count *n* = 41). Smaller colonies are those embedded in the agar.

**Table 1 microorganisms-08-02031-t001:** Results from transduction assays using two tetracycline- and five fusidic acid-resistant bacterial donors (MRSA and MRSP), and 27 MR- and MS- *S. aureus* and *S. pseudintermedius* recipients. Numbers represent transduction assays with the growth of more than two-fold higher bacterial colonies than negative control plates, compared to the number of replicate attempts. For tetracycline resistance, confirmation of successful transduction was made by PCR. For fusidic acid assays, all putative transductant colonies were subsequently shown not to carry *fusB* or *fusC*; mutations in *fusA* were identified by sequencing.

				Donor						
				*tet* *(K)*	*tet* *(M)*	*fusB*		*fusC*		
				MRSA (Human Hospital Environment)	MRSP (Canine Infection)	MRSP (Canine Infection)		MRSP (Canine Infection)		
				COL	1726	P0983	P1067	V1061	V1100	P1248
				CC8 (ST250)	ST261	ST621	ST1090	ST668	ST668	ST305
**Recipient**	**MSSP** **(Canine Infection)**	**221833**	**ST263**	1/10	3/10	Not Done				
		**287735**	**ST82**	0/3	1/10					
		**289869**	**ST54**	0/3	0/3					
		**289595**	**ST1903**	1/10	4/10					
		**289589**	**ST1907**	0/3	0/3					
		**289418**	**ST1905**	1/10	0/3					
		**289385**	**ST1906**	0/3	0/3					
		**V1273**	**ST1085**	Not Done		0/3	0/3	0/3	0/3	0/3
		**V0451**	**ST1091**			0/3	0/3	0/3	1/3	0/3
		**V0806**	**ST54**			0/3	0/3	0/3	0/3	0/3
		**P1361**	**ST1086**			0/3	0/3	0/3	0/3	0/3
		**P1351**	**ST21**			0/3	0/3	0/3	0/3	0/3
		**P1356**	**ST1092**			0/3	0/3	0/3	0/3	0/3
		**251648**	**ST71**			0/3	0/3	1/3	0/3	0/3
		**70361**	**ST1087**			0/3	0/3	0/3	0/3	0/3
	**MSSA** **(Canine Infection)**	**B019**	**CC15 (ST15)**	Not done	0/3	1/3	0/3	1/3	0/3	1/3
		**B021**	**CC15 (ST15)**		0/3	0/3	0/3	0/3	0/3	0/3
		**B027**	**CC15 (ST15)**		0/3	0/3	0/3	0/3	0/3	0/3
	**Restriction-deficient MSSA** **(Laboratory Strain)**	**RN4220**	**CC8 (ST8)**	4/10	0/3	0/3	0/3	0/3	0/3	0/3
	**Restriction-deficient MRSA** **(Laboratory Strain)**	**NE667 (hsdR mutant of JE2)**	**CC8 (ST8)**	Not done	0/3	0/3	0/3	0/3	0/3	0/3
	**MRSA** **(Human Infection)**	**JE2**	**CC8 (ST8)**		0/3	0/3	1/3	0/3	1/3	1/3
		**J220**	**CC8 (ST239)**		Not done	0/3	0/3	1/3	2/3	0/3
		**J225**	**CC8 (ST239)**			1/3	0/3	1/3	0/3	0/3
		**FPR3757**	**CC8 (ST8)**			2/3	0/3	0/3	0/3	0/3
	**MRSA** **(Human Carriage)**	**19B**	**CC22**		0/3	0/3	0/3	0/3	0/3	0/3
		**TW20**	**CC8 (ST239)**	Not done		0/3	0/3	2/3	1/3	0/3
	**MRSA** **(Human Hospital Environment)**	**COL**	**CC8 (ST250)**			0/3	0/3	0/3	0/3	0/3
**Total number transduction assays per antimicrobial**				115		300				
**Total plates with increased growth** **/total number of transduction assays**				7/52	8/63	4/60	1/60	6/60	5/60	2/60

MRSA: methicillin-resistant *S. aureus*; MRSP: methicillin-resistant *S. pseudintermedius*; MSSP: methicillin-susceptible *S. pseudintermedius*; MSSA: methicillin-susceptible *S. aureus*.

**Table 2 microorganisms-08-02031-t002:** Number of transductant cells/mL following successful transduction assays for *tet(M)* and *tet(K)*. Cell numbers are derived from colony counts following transduction assays incubated at 37 °C for 48 h on LK agar containing 30 mg/L tetracycline.

Donor			Recipient		Number of Successful Transduction Assay Replicates	Median (Range) Transductant Cells/mL
Bacterial Type	Isolate	Tetracycline Resistance Gene	Bacterial Type	Isolate		
**MRSP**	**1726**	***tet(M)***	MSSP	221833	3/10	1105 (250–1510)
				287735	1/10	1535
				289595	4/10	92.5 (25–160)
**MRSA**	**COL**	***tet(K)***	MSSP	221833	1/10	1475
				289595	1/10	65
				259418	1/10	40
			RM-def MSSA	RN4220	4/10	62.5 (25–995)

MRSA: methicillin-resistant *S. aureus*; MRSP: methicillin-resistant *S. pseudintermedius*; MSSP: methicillin-susceptible *S. pseudintermedius*; RM-def MSSA: restriction-modification system deficient methicillin-susceptible *S. aureus*.

**Table 3 microorganisms-08-02031-t003:** Fusidic acid minimum inhibitory concentrations (MIC) and mutations in *fusA* (including the novel position F88L) after exposure of MRSA, MSSA, and MSSP to subinhibitory concentrations of fusidic acid. PCR confirmed species and methicillin-resistance as the same as the original recipient isolate.

Staphylococci	Original Recipient (Recipient MIC [mg/L])	Mutant MIC (mg/L)	Amino Acid Substitution	Nucleotide Substitution
**MSSP**	251648 (0.06)	32	H457Y	CAC → TAC
**MSSA**	B019 (0.06)	32	H457Y	CAC → TAC
		32	F88L	TTC → CTC
**MRSA**	TW20 (0.06)	32	F88L	TTC → TTA
	J220 (0.06)	>64	H457N	CAC → AAC
		32	H457Y	CAC → TAC
	J225 (0.06)	32	H457Y	CAC → TAC
		64	H457N	CAC → AAC
		64	H457N	CAC → AAC
	FPR3757 (0.06)	64	H457L	CAC → CTC
	JE2 (0.06)	32	H457Y	CAC → TAC

MSSP: methicillin-susceptible *S. pseudintermedius*; MSSA: methicillin-susceptible *S. aureus*; MRSA: methicillin-resistant *S. aureus*.

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
