# Peer review of "Genes on the Move: In Vitro Transduction of Antimicrobial Resistance Genes between Human and Canine Staphylococcal Pathogens"

_microorganisms, 2020, doi:10.3390/microorganisms8122031_

Round 1
Reviewer 1 Report
General comments:
This paper provides an insight into the mechanisms of antimicrobial resistance acquisition and genetic transfer between two different Staphylococcus species carried and or infecting human and dogs, and demonstrates the potential risk of transmission of MT Staphylococcus between pets and humans. It is particularly interesting as this field has been relatively less explored compared to the mass of publications about antimicrobial resistances in agriculture and farm animals. It is an interesting and well written paper, which adds important data to the current knowledge.There are only minor revisions to be done to polish this paper (wording or clarifications).
Specific comments:
L25: “MSSP”: full name needed
L48: remove the first carriage, co-carriage is sufficient
L51: “SCCmec”: full name needed
L73-75: this phrase is a bit confused, please rephrase it
L79: “Low-level FA resistance” should be plural (levels) and of added
L102-103: if you give one number (n=1), then you need to give the others too, should also refer to Table 1.
L104: “globally” => replace with worldwide
L104-106: “confirmed by PCR”: should indicate following method described in…
L109-110: donor isolates for FA are 5 MRSP isolates, why no MRSA?
L124: “stocks were purity plated”: needs to be reworded
L129: prophage should be plural
L168: “amplified and sequenced” => previously described in…
L183: “MSSP”: full name for first time appearance
L195-196: Additional assays were performed for the successful pairings with tetracycline, why not with FA?
L196: Please clarify “significant growth”. Why is it different compared to tetracycline?
L207: Table 1: There is no need to include the full names MRSA , MRSP, etc. here in the title as you indicate them further in L212.
Table 1: Why are the MSSP recipient isolates tested with tetracycline donors and FA donors different?
Why have some of the other recipient isolates (MSSA and MRSA) been tested with only some donors and not others?
Restriction-deficient MRSA (laboratory strain) NE667: should it be rather MSSA?
Total number TD => full name for TD
L242-245: “from the human pathogen S. aureus into dog-adapted S. pseudintermedius” : the other way too? S. pseudintermedius could also transfer resistance genes to S. aureus?
L250-251: “no transduction occurred” => this phrase needs to be clarified, as it is unclear if you discuss here the results for tetracycline or for FA
L257: "the low reproducibility... surprising": again you need to specify this is related to tetracycline, for easier reading
L273-277: Any explanation why this difference between in vivo and in vitro? It could be expected to see the contrary (HGT higher in vitro than in vivo)
L288-293: The conclusion could be more developed: What would be the next steps? Should there be a greater surveillance of the use of antimicrobials in veterianry settings to prevent multi-resistance acquisitions? Should drugs such as FA be strictly reserved to the human use, and alternative protocols developed for pets? What about other pets: cats, rodents?
Reviewer 2 Report
Methicillin -resistant Staphylococcus aureus (MRSA) is a common pathogen known for its ability to cause skin and soft tissue infections, often resulting in continued administrations of antibiotics in a hospital setting for complete resolution of the disease. MRSA is widely known for its ability to acquire resistance to a plethora of antibiotic regimens, especially after prolonged treatments. Likewise, other members of the staphylococci that inhabit other environmental niches also have a propensity to acquire and disseminate antimicrobial resistant determinants between and withing species. One particularly relevant staphylococci is Methicillin -resistant Staphylococcus pseudintermedius (MRSP) which has emerged as a highly drug resistant pathogen that colonizes (and infects) dogs and has many similarities to MRSA. Importantly, MRSA and MRSP have been shown to co-colonize -at least transiently- in humans and dogs, which naturally leads to the question of horizontal transfer between these two species antimicrobial resistant determinants, which if occurs, can have major implications in the evolution and spread of antimicrobial resistant pathogens.
In the submitted manuscript by Frosini et al., the authors demonstrate that horizontal transfer of antimicrobial resistant determinants (i.e. tetracycline resistant markers tet(M) and tet(K)) can occur in vitro through endogenous bacteriophage mediated transduction between clinical strains of Staphylococcus aureus and Staphylococcus pseudintermedius using lysate derived from UV-mediated prophage induction of tetracycline resistant donor strain incubated with tetracycline sensitive recipient strain. The transmission of tetracycline resistant determinants from MRSP donor strain to MRSA recipient strain was not observed, as well as the transfer of fusidic acid resistant determinants between either of these strains; However, fusidic resistant strains did arise through suppressor mutations in previously known and novel sites within the fusidic resistant gene fusA.
Overall the manuscript is well written and thoughtfully designed, and should be considered for publication following the addressment of several major and minor points
Major point
Lines 174-177: The bacterial phage titers, derived from donor strains, could not be determined since the RN4220 (restriction deficient strain control) did not show any plaques after incubation with phage lysate derived from donor strain. This is very peculiar and perhaps concerning since transduction did occur between various recipient strains (with presumably fully functional restriction modification systems) and lysate from donor cells to allow the transmission of tet resistant determinants. Maybe the authors could add more commentary to this and possibly use a different strain to confirm results and test for titers. Deficiency in bacterial phage titer could be responsible for the lack of transmission of tet resistant determinants from MRSP to MRSA and for the lack of fusidic resistant gene transfers.
Lines 273-277: In the discussion the authors mention that interspecies horizontal gene transfer maybe greater in vivo than what’s observed in the laboratory, and then goes on to cite examples where this has been reported in the literature. I think it would behoove the authors to elaborate on why this may be so since all their data is strictly in vitro. I think it would strengthen the manuscript if they can give tangible reasons why transfer in vivo could be more prevalent than in vitro, thus giving more physiological credibility to their conclusions.
Minor points
Line 52: add relevant citation after sentence (maybe cite a review article if available)
Line 53: add relevant citation after sentence (maybe cite a review article if available)
Line 54: remove italic (i.e. for to for)
Lines 80-83: Please reword the sentence – its very hard to follow (maybe break up into two separate sentences)
Line88: add relevant citation after sentence
Round 2
Reviewer 2 Report
Based on the authors response -and new edits to the manuscript- I would recomend publication.